# A CA–Markov-Based Simulation and Prediction of LULC Changes over the Nyabarongo River Basin, Rwanda

**Aboubakar Gasirabo** [1,2,3], **Chen Xi** [1,2,*], **Baligira R. Hamad** [3,4] **and Umwali Dufatanye Edovia** [1,2,3]

1   State Key Laboratory of Desert and Oasis Ecology, Xinjiang Institute of Ecology and Geography, Chinese Academy of Sciences, 818 South Beijing Road, Urumqi 830011, China; gasirabo2@mails.ucas.ac.cn (A.G.)
2   University of Chinese Academy of Sciences, Beijing 100049, China
3   Faculty of Environmental Studies, University of Lay Adventist of Kigali (UNILAK), Kigali P.O. Box 6392, Rwanda
4   Rwanda Energy Group REG, Kigali P.O. Box 537, Rwanda
*   Correspondence: chenxi@ms.xjb.ac.cn

**Abstract:** Over the past few decades, the growth of population and the development of the economy have had a significant impact on the way land is used and covered (LULC) in the Nile Nyabarongo River basin. However, there is limited knowledge about the patterns of land use and the mechanisms that drive changes in these patterns because of human activities. Therefore, it is crucial to examine how land use and cover are shifting in this area, identify the factors responsible for these changes, and forecast future patterns. This study sought (1) to evaluate the changes in LULC from 1990 to 2020 and (2) to predict future fluctuations until 2060. By analyzing the LULC data for the years 1990, 2000, 2010, and 2020, this study simulated the future LULC patterns of the area for the next 30 years using the LULC transition matrix and the Markov chain model. The study classified the LULC into five categories: forestland, grassland, cropland, settlement, and water. The results revealed that there will be significant changes in the LULC of the study area from 2030 to 2060. The forest area is projected to decrease by 801.7 km, 771.6 km, and 508.2 km, while the cropland area will expand by 6307.2 km, 6249.2 km, and 6420.6 km during this period. The grassland area will experience a small increase of 761.1 km, 802.4 km, and 859.1 km, and the settlement area will also grow by 355.2 km, 407.4 km, and 453.2 km. In contrast, the water area will decrease by 55.9 km, 50.5 km, and 40 km. The ongoing pattern of LULCC is expected to persist over the next three decades, with an increase in cropland area and grassland. This study's findings can provide valuable insights for land use planners and water resource managers in developing fair land use and water resource management policies for the entire region, enabling them to make well-informed decisions.

**Keywords:** LULC; GIS; CA–Markov; Nyabarongo River; Rwanda

## 1. Introduction

LULCC refers to the alterations made by humans to the land surface on Earth [1]. These modifications have major implications for the ecosystem on local, regional, and international scales. It has become an international concern because of the effects it has on our surroundings [2,3]. To address this issue, the use of geospatial models and open-access RS data has been shown to be an effective instrument for monitoring and managing the status and changes in LULC. This approach can greatly contribute to conservation efforts and sustainable land management [4]. The improvements in earnings and population expansion have directed to the extension of cities and city areas, which is an unavoidable consequence of human activity. The methods of land usage in these cities' areas are affected by several human and natural variables such as industrialization, modernization, globalization, marketization, and administrative [5–7]. It is vital to keep in mind that the quick expansion of cities and urban growth have significant impacts on the economy of societies. This issue is prevalent worldwide and is often accompanied by poor planning,

resulting in various negative consequences such as deforestation, loss of cropland, and conversion of grassland into urban environments [8–10]. In the coming years, the world will encounter a series of interconnected land-use challenges caused by the expected rise in demand for goods and services, placing strains on limited land resources [11]. The awareness needed to make decisions about ecological management and long-term sustainable development will, therefore, depend on data on LULC variations. As a result, these data are crucial for evaluating the studies and discussions on the current worldwide change scenario. Due to the predicted increases in demand for products and services from restricted land resources, the world will soon face new interrelated land-use concerns [12].

Current studies [13,14] have shown a notable 20% rise in emissions from African waterways, which is between 47 and 57 million tons of carbon dioxide per year through 1990. Specifically, fluxes of greenhouse gases brought on by the degradation of organic soils in sub-Saharan Africa's waters (apart from South Africa) account for roughly 25% of the region's overall fossil fuel pollution. In the region, researchers have studied LULC in Nile Nyabarongo, and they have primarily focused on the following three aspects, namely policy, soil properties, and land-use transfer (land-use/land-cover change). In terms of policy, researchers believe that more smallholder land can improve grain output to some extent and enhance food security [15,16]. Franchi, et al. [17] argued that smallholders play a dominant role in rural areas in Rwanda because they acquire their land mostly through borrowing.

Numerous modeling techniques, such as the land-use conversion and its effects (CLUE) (Das et al.) [18] and future land-use simulation (FLUS) (Lin et al.) [19], have been applied for the purpose of predicting the spatial and temporal variations in LULC. Among these models, the CA–Markov models [20,21] have been popular because of their aptitude in integrating the Markov chain theory with cellular automata (CA) fundamentals to create a robust and comprehensive approach that considers the complexity of the landscape [22,23]. Furthermore, the method provides valuable insights into complex spatial systems. Various researchers have utilized the CA–Markov model in various studies with success. For instance, Ruben et al. [24] and Kisamba and Li [25] employed the CA–Markov model to forecast changes in land cover and to advise on land-use policies and future planning. In addition, in Zimbabwe Sibanda, Ahmed and Environment [13] simulated projected LULC in the Shashe subcatchment using the CA–Markov model and assess its impact on the water area. Matlhodi et al. [26] evaluated the GEOMOD and CA–Markov models for forecasting future LULCCs in the Nepali Phewa Lake drainage. Their findings showed that the CA–Markov model performed better in prominent future LULC circumstances than GEOMOD. We conducted a study on LULC prediction for the years 2030, 2040, 2050, and 2060. This study, therefore, applied supervised classification and CA–Markov model (1) to assess the current changing LULC patterns in the Nile Nyabarongo River's LULC and (2) to simulate the future LULC in the Nile Nyabarongo River. It is worth noting that there is only one study in Rwanda that has used the CA–Markov model to simulate LULC on a national scale [27]. Thus, our research will be one of the first to apply this model to forecast future LULC patterns in the Nile Nyabarongo River, Rwanda. The main goals of this study are to measure and track long-term changes in LULC using RS methods and GIS. Additionally, we aim to forecast the location and distribution of various land-use types using the CA–Markov model.

Researchers have examined the historical LULC data in the Nile Nyabarongo River basin, assessed the changes in LULC throughout various time periods, and determined the causes of these changes. Predictions regarding future land use in the area are then made using a full understanding of the driving causes behind LULCCs, as well as demographic, topographic, and locational considerations. The outcomes of research are anticipated to contribute to the improvement of natural resource management and the living conditions in the area.

## 2. Materials and Methods

### 2.1. Description of the Study Area

The Nyabarongo River basin is regarded as the Nile River's source and is situated in Rwanda (Figure 1) [28]. It is subdivided into three subbasins, namely, the Nyabarongo upstream subbasin in the south, Nyabarongo downstream subbasin in the east, and Mukungwa subbasin in the north.

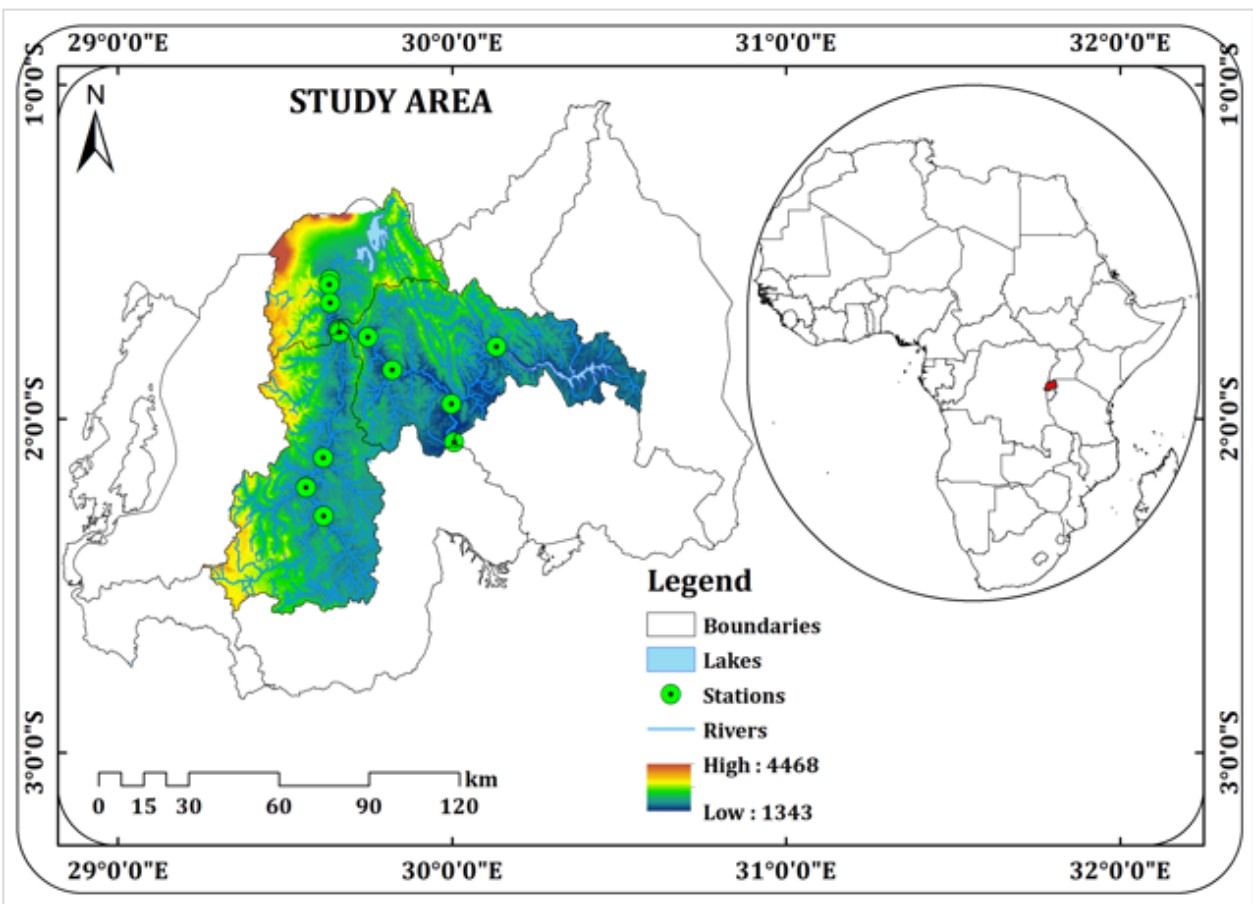

**Figure 1.** Geospatial location map of the study area.

The biggest and most important river in Rwanda is the Nyabarongo River, covering a total area of 8416.21 km$^2$. It flows more than 300 km from its origin in western Rwanda toward Lake Rweru in southeastern Rwanda along the border with Burundi [29]. The Akanyaru River is the main tributary of the Nyabarongo and originates alongside the border between Rwanda and Burundi in the mountains of Nyungwe National Park on the Congo–Nile separate. The Nyabarongo River then proceeds on its course eastward through the lowlands of Bugesera-Gisaka in southeast Rwanda, passing through watery valleys and numerous lakes [30]. As it passes through Kigali city, it is fed by the Mwange, Rusine, and Marenge rivers, among others, on its upstream stretch. Additionally, it draws water from other rivers in Kigali's metropolitan areas, including as the Rwanzekuma, Ruganwa, Mpazi, and Yanze rivers [28]. Human-made activities like farming, sugar cane farms, mineral extraction, farming, and floriculture are the main sources of pollution in the Nyabarongo River [28].

### 2.2. Datasets

To successfully utilize satellite imageries from various times, it is important to consider factors such as a cloud-free spatial coverage, high spatial and spectral resolution, and images acquired on the same date for the year [31]. The downloaded images in this study

were projected using the Universal Transverse Mercator (UTM) method and georeferenced using the WGS84 datum and ellipsoid [32]. Preprocessing operations were carried out using the image analysis tools in ArcGIS 10.8 and ENVI 5.3 software [33]. The study utilized various Landsat datasets (Table 1) based on the mentioned criteria, relevant literature, and the widespread use of Landsat images for land-use studies. Specifically, four remotely sensed satellite images with a 10-year interval were employed to investigate the dynamics of LULC (Table 1). The Landsat image scenes were obtained from the United States Geological Survey (USGS) website, a freely accessible data portal (http://earthexplorer.usgs.gov/ (accessed on 26 June 2021)). These data (Table 1) were partially preprocessed and had undergone radiometric, geometric, and atmospheric corrections, which, thus, eliminated the need for image registration. In order to aid classification, particularly because of the presence of waters in low lying areas, the Shuttle Radar Topography Mission (SRTM) was used to acquire a 30 m digital elevation model (DEM) of the terrain.

**Table 1.** RS imageries used in this study.

| Year | Path/Row | Acquisition Date | Sensor Type | Spatial Resolution (m) | LULC Name | Source |
|------|----------|-----------------|-------------|------------------------|-----------|--------|
| 1990 | 173/61 | 25/07/1990 | Landsat 5 TM | 30 | 1990 LULC | USGS |
| 2000 | 173/62 | 27/09/2000 | Landsat 7 ETM | 30 | 2000 LULC | USGS |
| 2010 | 172/61 | 22/07/2010 | Landsat 8 OLI | 30 | 2010 LULC | USGS |
| 2020 | 172/62 | 27/09/2020 | Landsat 8 OLI | 30 | 2020 LULC | USGS |

### 2.2.1. Field Data

In order to identify LULC classes in various regions of the Nile Nyabarongo basin, we carried out a field survey using random sampling. We compared the unique spectral properties of each LULC class on the maps, then compared the identified LULC classes with the matching types observed in satellite images.

For each season in each period, an inventory of unique training sites was generated. For 2010–2020, training sites were collected through field visits using handheld GPS units. For 2000, training sites were selected following on-screen digitization through visual interpretation of seasonal Landsat data and Google Earth imagery. For 1990, due to the unavailability of Google Earth imagery, only seasonal Landsat imagery and expert knowledge derived from other periods were used to select training sites. During the collection of training samples of crop land use, only cropped or noncropped lands were identified for seasonal data classification. To minimize any kind of errors related to mixed pixels, homogenous polygons were selected as training sites. Finally, from the training polygons, 200 sample points for each class were generated following a stratified random sampling to remove any kind of bias arising from under/over-representation of a particular class. Additionally, independent samples (~25 sample points for each LULC class) were collected for accuracy assessment, including information on cropping practices along with other LULC classes.

The projected LULC categories that were identified through both the field survey and satellite images include forests, grasslands, croplands, settlement, and water. This information was used to classify and create a map of the five different LULC types (Table 1).

### 2.2.2. Evaluation of Classification Accuracy

Assessing the reliability and accuracy of image classification methods involves an important evaluation process [34,35]. Previous studies [36,37] have recommended an accuracy level above 90% for LULC classification to ensure excellence and reliability. Model validation and accuracy play a crucial role in conducting a reliable study on LULC [38]. This study applied the kappa and overall accuracy statistics to evaluate the classification maps' effectiveness. To validate the results, GPS locations, Google Earth images, and satellite maps were employed. We generated a confusion matrix that merged the classified LULC map with ground-based reference information. Various measures such as the producer precision, user reliability, overall precision, and kappa efficiency were determined [39]. The

overall precision was determined by the percentage of appropriately categorized pixels in the error matrix, and the kappa coefficient demonstrated how well the classified map matches with the data used for reference [40].

The aforementioned equations were used to calculate these two measurements:

$$\text{OverallAccuracy} = \frac{\sum_{i=1}^{r} Xii}{D} * 100 \tag{1}$$

where $D$ is the overall number of pixels in the matrix table, $r$ is the number of rows in the matrix, and $x_{ii}$ indicates the total number of pixels correctly recognized in row $i$ and column $i$.

$$\text{Kappa Accuracy} = \frac{D\sum_{i=1}^{r} x_{ii} - \sum_{i=1}^{r} (x_{i+} * x_{+i})}{D^2 - \sum_{i=1}^{r} -(x_{i+} * x_{+i})} \tag{2}$$

where $r$ is the matrix's quantity of rows, $x_{ii}$ denotes the total number of pixels in row $i$ and column $i$ that were correctly classified, and $x_{i+}$ and $x_{+i}$ denote the marginal sums of row $i$ and column $i$, respectively.

The use of kappa indexes for the calculation determines the overall achievement rate, and it delivers an understanding of the real factors in the strength or weakness of the results. When 75% ≤ kappa < 100, the result maps are in a high level of agreement; if 50% ≤ kappa ≤ 75%, the result maps are in a medium level of agreement; and if kappa ≤ 50, the result maps are in a poor agreement [40,41]. Therefore, to know the accuracy of the CA–Markov model in simulating future LULC conditions, the model was confirmed by Wang and Zheng [42] after simulating the 2020 LULC situations using the 1990 and 2020 classified images. The kappa index of agreement ($K_{IA}$) (Wan, et al.) [43], such as kappa for no information ($K_{no}$), kappa for location ($K_{location}$), and kappa for standard ($K_{standard}$) (Matlhodi et al.) [26,44], evaluated the agreements of the two maps (actual and simulated 2020) using the CROSSTAB module in IDRISI. In addition, comparisons of the simulated and the actual area of each LULC class were also performed using the validate module. Hence, the kappa index is acceptable; the LULC from 2030 to 2060 can be predicted.

*2.3. Methods*

2.3.1. LULCC Detection

LULCC analysis is essential in identifying the specific shift occurring in different land-use classes [45]. The map of the LULC identification of changes was utilized to evaluate and examine the temporal changes of LULC within the designated area.

The equation mentioned below was employed to determine the extent of change in each class:

$$C_t = Y_I - T_i \tag{3}$$

where $C_i$ indicates the class $i$ change in extent, $T_i$ the base image, and $Y_i$ the most recent image. Using the subsequent equation, we determined the percentage change for each LULC class:

$$P_i = \frac{Y_i - T_i}{T_i} * 100 \tag{4}$$

where $P_i$ symbolizes the class $i$ shift in percentage terms, $T_i$ the initial image, and $Y_i$ the most recent image.

2.3.2. LULC Prediction Using the CA–Markov Chain Model

Cellular automata and Markov chain, multi-criteria, and multi-objective land allocation (MOLA) approaches are all combined in the modeling methodology known as the CA–Markov chain. The objective of this approach is to forecast future trends and LULC alteration properties. The CA model takes into account the uncertainty that arises from various factors, including the correlation between model components, the model's design, and the accuracy of the data used as input [46,47], while the CA–Markov approach focuses on how cells react locally, taking into account their unique spatial and temporal character-

istics. It utilizes computational capabilities that are suitable for dynamic simulation and visualization. One of the key applications of the CA–Markov chain model is to analyze transitional probabilities of different LULC classes in various time spans [48]. By studying these transitional probabilities, we gain insights into the drivers of land-use changes and how they may continue in the future. This understanding allows LULC characteristics and their possible effects on the environment, natural resources, and landscape creation to be forecast [49]. The CA–Markov model, which takes advantage of the Markov model's nearness and the benefits of CA, has been demonstrated to be a successful tool for simulating changes in land use in previous research [50–52]. We can determine the prospective spatial distribution of transitions using this model [22,53]. Throughout the process, the following steps were followed: the base map, LULC maps from 1990, 2000, 2010, and 2020, as well as a transition compatibility imagery, were utilized to create the LULC maps from 1990, 2000, 2010, and 2020. The transition feasibility image and the transition probability image were processed using the IDRISI Terrset program. The transition probability map for the years 2000 to 2010 was subsequently created to replicate the LULC map for 2020 and the Markovian transition estimator technique was used to model the LULC maps for 2030, 2040, 2050, and 2060 for the years 2010 to 2020. To determine the transition compatibility image, the limitations and factors of the multi-criteria evaluation (MCE) module were taken into consideration [54,55]. The CA–Markov model uses time to reveal the patterns and factors that lead to future adjustments.

Thus, CA model can be described as follows (Equation (5)):

$$S(t, t+1) = f(S(t), N) \tag{5}$$

where the N operate is the state frequency at any time, and S(t + 1) is the system status at the moment of (t, t + 1). This model is frequently utilized to conduct simulation and ecological modeling and LULC monitoring and to predict future stability and change in land use in a specific area. Future changes in LULC are predicted using the following formula:

$$S(t, t+1) = P_{ij} * S(t) \tag{6}$$

where $P_{ij}$ is the transition probability matrix in a state, which is calculated as follows, S(t) is the system status at time t, and S(t + 1) is the system status at time t + 1:

$$= \left\| P_{ij} \right\| = \left\| \begin{matrix} P1,1 & P1,2 & P1,N \\ P1,1 & P2,2 & P2,N \\ PN,1 & PN,2 & PN,N \end{matrix} \right\| (0 \le P_{ij} \le 1) \tag{7}$$

where *P* is the transition probability, $P_{ij}$ is the likelihood a specific state will most likely remain in existence at any given time, while PN is the likelihood that it will shift from one state to another in the future. Compared to a low transition, a high transition has a chance that is near to 1 [56].

We performed a Markov chain analysis to generate the transition matrix of the LULCC and the probabilities of change from 1990 to 2000, 2000 to 2010, and 2010 to 2020. The transition matrix serves as the basis for projecting future LULCC dynamics. However, the Markov chain model is not explicit spatially. It lacks the scientific explanation of the processes of change and neglects the spatial distribution of LULC, which is exceptionally significant in simulating land-cover patterns [57]. The cellular automata (CA) model is also widely used for LULC prediction because of its spatial capacity to alter and control processes of complex distributed systems. The CA model comprises the cell, cell space, neighbor, time, and rule. The model describes the new pattern of LULC, considering the state of previous neighborhood cells [58,59]. The distance between the neighbor and the cell defines the weight factor of changing to a particular land cover. The weight factor of the change to a specific land cover is determined by the distance between the cell and the neighbor. The weight factor was then combined with the transition probabilities to estimate neighborhood cells' conditions so that change prediction is not only based on

a random decision. The cellular automata (CA) model considers Markov's previous or current state of LULC and utilizes the neighborhood cells' conditions for its transition rules [60]. It provides a suitable environment for dynamic modeling in GIS and remote sensing because of its analytical engine [61]. However, the CA model has limitations in defining transition rules and modeling structures. Therefore, the combination of different empirical and dynamic models, such as the CA–Markov model, is vital in achieving a dynamic LULC spatial modeling [62].

The study used the land-change modeler (LCM) in IDRISI-TerrSet geospatial monitoring and modeling software to monitor and simulate the potential LULC change dynamics from 2020 to 2060. The procedures used to run the CA–Markov in LCM comprised the following procedures. The first stage involved running three different models using the land-cover maps of 1990–2000, 2000–2020, and 1990–2020. This procedure was used to generate the transition probability matrix (TPM), transition area matrix (TAM), and transition suitability maps (TSM). The TPM is obtained by cross-tabulating two multitemporal images. It contains the probability of each land-cover class changing to another over a predetermined time. The TAM contains the estimated number of pixels that might change over a specific number of time units from a particular land-cover class to another, while the conditional probability map expresses the probability of each land-cover-class pixel having a place with the assigned class after a specific time [63]. The transition suitability maps (TSM) were generated for the five land-cover classes used in the study (i.e., forestland, grassland, cropland, settlement, and water).

The model of different historical LULC scenarios of 1990–2000 and 2000–2020 was used to produce the transition probability matrix of periods 1 and 2. The second stage involved using a standard contiguity filter of 5 × 5 to define each cell's neighborhoods and generate the spatially explicit weighing factors. After calibrating the model, we used the scenario-bound approach to simulate the potential LULC pattern in the final stage. The process involved using the classified land-cover maps of 1990 and 2000 to calibrate and optimize the Markov chain algorithm. The earliest year (i.e., 1990) was used as time 1, while the later year (i.e., 2000) was used as time 2. The transition probabilities between time 1 and 2 were used to simulate the LULC pattern in 2020. The study validated the CA–Markov model to determine the accuracy of the 2020 prediction. The process requires a statistical tool to differentiate between location errors and quantities' errors in analyzing the similarities between two images [64]. We employed the kappa statistics index to determine the level of agreement between the projected and the actual land-cover map of 2020. The result indicated an acceptable kappa index, signifying a reliable LULC modeling and prediction. Therefore, the 2020 classified land-cover map was used as a base map to forecast the potential LULC in 2030, 2040, 2050, and 2060.

The Markov chain calculates the alteration in land area from the current year to the anticipated year. It produces potential LULC possibilities, which also provide policy makers with a greater understanding of how to adapt to the factors influencing changing vegetation [65]. This model illustrates the dynamics of LULC, vegetation, the expansion of urbanized areas, and watershed planning modeling. It is essential for planning and developing various land-use policies that will promote appropriate LULC management [66]. (see Figure 2).

The model accurately predicts changes in LULC by considering the factors that drive land-use transition, concurrence among various land-use forms, and other pertinent considerations (such as transportation routes, rivers, and cities). This model is particularly effective at simulating the LULC pattern with scattered patches. Furthermore, studies have demonstrated that the CA–Markov model outperforms other models, such as the conversion of land use and its effects (CLUE), future land-use simulation (FLUS) model, and artificial neural networks–cellular automata (ANN-CA), in terms of accurately simulating LULC and landscape patterns [67,68]. (see Figure 3).

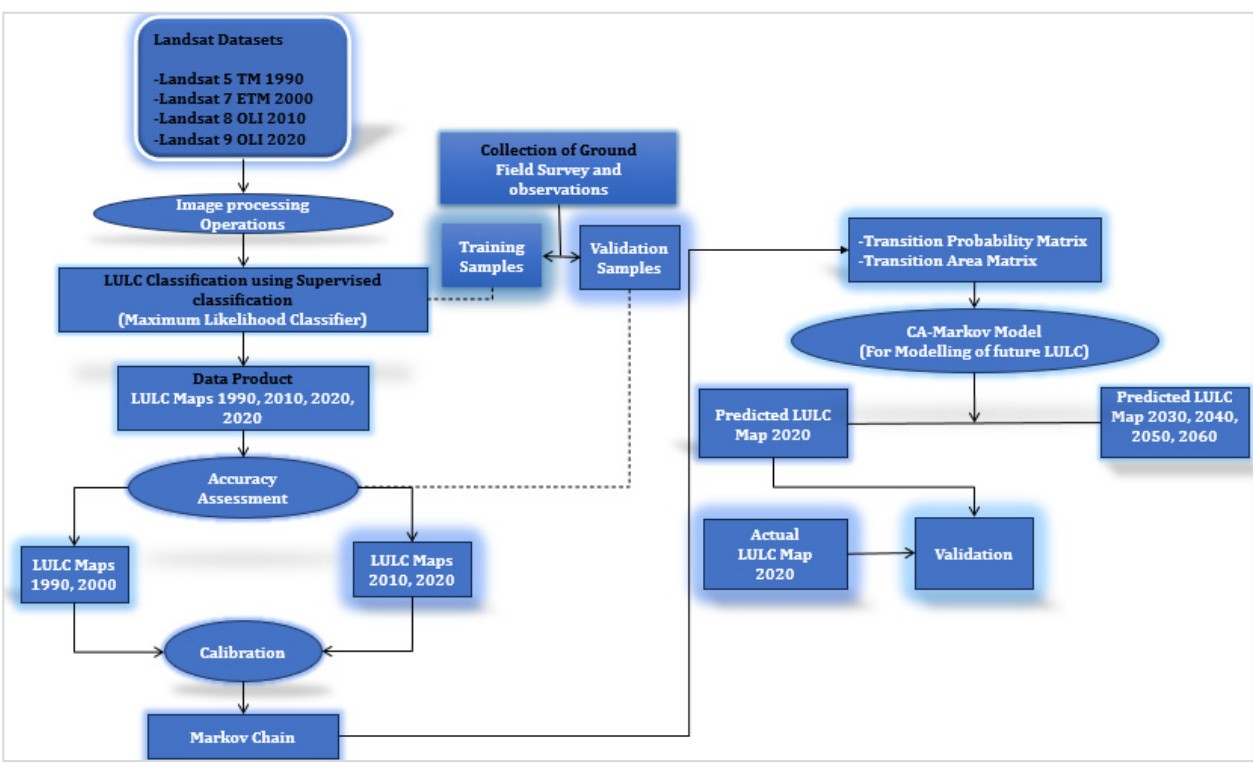

**Figure 2.** Research methodological flowchart.

### 2.3.3. Model Calibration and Validation

When estimating without having exposure to datasets, potential future alterations for evaluating the accuracy of the forecasts, model calibration, and validation are essential parts of the modeling analysis. In the study by Nath et al. [67], they used chi-square (x2) test statistics to ascertain whether the projected LULC images are accurate as compared to the actual LULC for 2020. However, this comparison alone may not provide a complete evaluation of the LULC categories' spatial distribution at the study area. The researchers used the kappa index agreement, an additional sophisticated approach, to address the issue [69]. This technique involved calculating the kappa metrics for location, quantity, and no information ($K_{no}$, $K_{location}$, and $K_{quantity}$, respectively) to distinguish between the anticipated and real visuals. These kappa metrics, which distinguish errors of quantity and spatial errors were utilized to evaluate the precision of the CA–Markov model. Higher ratings indicate a higher level of agreement. The results might vary between 0 to 1. According to Beroho et al. [70], kappa values ≤ 0.5 kappa = 1 reflects absolute agreement, incredible agreement is indicated by a kappa of 0.5, medium agreement by a kappa of 0.75, high agreement by a kappa of 0.75, and incredible agreement by a kappa of 1. (see Tables 2–5).

**Table 2.** Definition of kappa indices.

| Kappa Index | The Kappa Index Consensus Specifications |
| --- | --- |
| $K_{no}$ | Without the ability to precisely set the quantity of location, this indicator shows the proportion of correctly identified cases against the anticipated percentage of correctly designated cases from simulation. |
| $K_{location}$ | Depending on a specific place, evaluates the consistency within the generated map and the categorized maps. |
| $K_{standard}$ | The ratio between the proportion correctly allocated and the proportion that is accurate by probability |

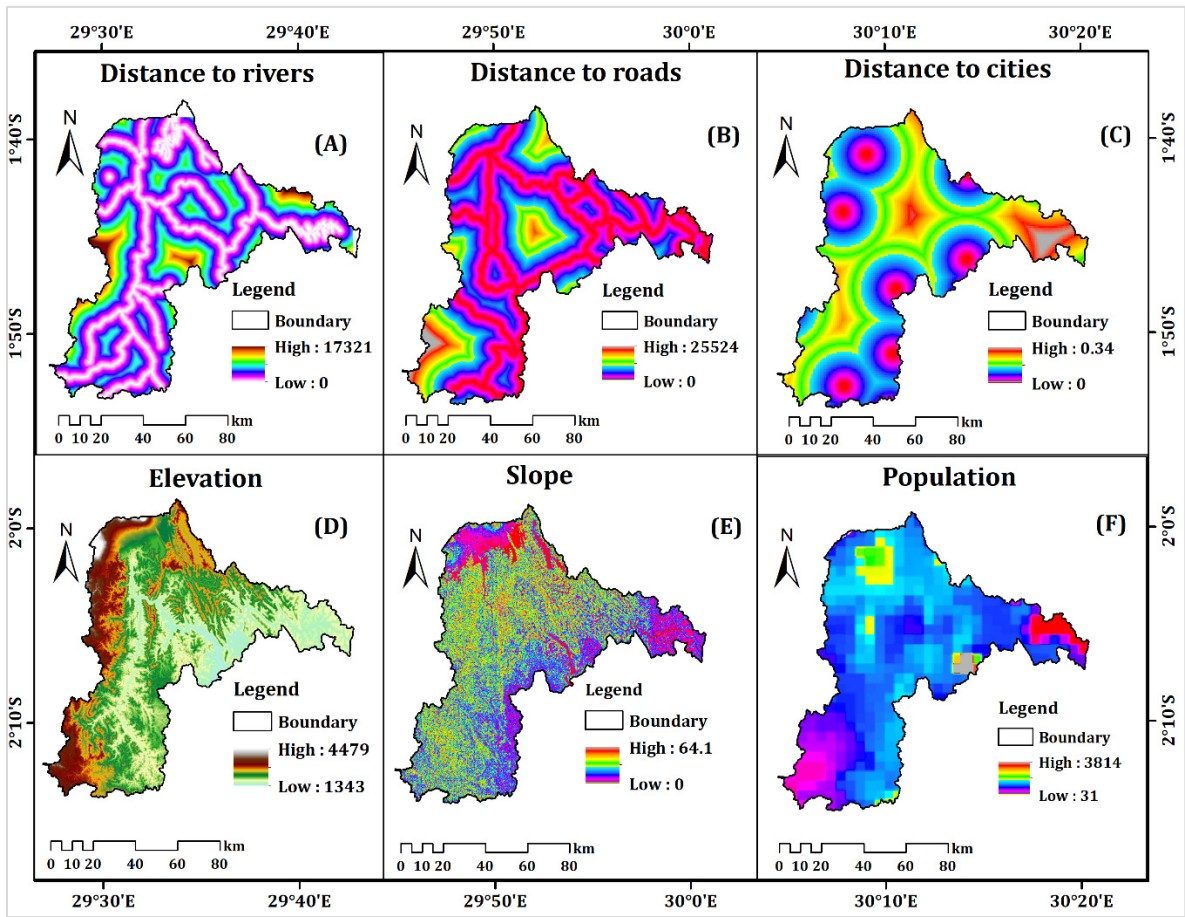

**Figure 3.** Factors driving LULCC: distance to rivers (**A**); distance to roads (**B**); locations of major cities (**C**); elevation (**D**); slope (**E**); and population (**F**).

**Table 3.** Accuracy assessment of the simulated land-use image of 2020.

| Index | Value |
|---|---|
| $K_{no}$ | 0.956 |
| $K_{location}$ | 0.939 |
| $K_{standard}$ | 0.629 |

**Table 4.** Outcomes of the validation assessment (agreement/disagreement factor values) of two pictures.

| Agreement/Disagreement | Value | Value (%) |
|---|---|---|
| Agreement chance | 0.16 | 16.5 |
| Agreement quantity | 0.38 | 38.6 |
| Agreement grid cell | 0.43 | 43.1 |
| Disagreement grid cell | 0.03 | 0.3 |
| Disagreement strata | 0.00 | 0.0 |
| Disagreement quantity | 0.01 | 0.1 |

**Table 5.** Verification of the change forecast using existing and anticipated 2020 data.

| LULC Classes | Area of the Predicted LULC 2020 (%) | Area of the Actual LULC 2020 (%) | $(P - A)^2/A$ |
|---|---|---|---|
| Forestland | 55.2 | 53.6 | 0.07 |
| Grassland | 7.4 | 8.2 | 0.05 |
| Cropland | 32.2 | 31.1 | 0.00 |
| Settlement | 3.7 | 6.9 | 0.60 |
| Water | 1.5 | 1.5 | 0.03 |
| Total | 100 | 100 | 0.75 |

The CA–Markov model was determined to be appropriate for simulating the LULC map of 2060 utilizing the transition probabilities from 2010 to 2020 and the categorized map of 2020 as the basis based on its validation utilizing kappa indices. Thus, the model was considered suitable for this purpose. The metrics were calculated using the equations below:

$$K_{no} = (W(v)N(n))/(S(s) - N(n)) \tag{8}$$

$$K_{location} = (W(v)N(n))/(S(v) - N(v) \tag{9}$$

$$K_{standard} = (W(v)N(n))/(S(s) - N(v)) \tag{10}$$

where $W(v)$, $N(v)$, and $S(v)$ generate moderate grid cell-level information, $N(v)$, and $S(v)$ and $S(s)$ define perfect grid cell-level information throughout the terrain.

$$X^2 = \sum \frac{(P - A)^2}{A} = 0.75$$

The percentage share of each LULC class utilized for model validation is represented by the data in the second and third columns, respectively, as in P and A.

The Terrset's LCM module was utilized to assess the performance of the model [71,72]. An evaluation was conducted to determine the agreement or disagreement components (Table 4), which were further classified into two categories: 0.02 (representing errors related to allocation/disagreement in grid cells) and 0.01 (representing errors related to quantity/disagreement in quantity). Consequently, the findings indicate that allocation errors, rather than problems with quantity, were the main cause of the disagreement between the observed and actual data. (see Figure 4).

The LULCCs in various years were simulated using kappa variations as a measure of accuracy, influenced by the demand for LULC in the future (Figure 5). Figure 6 displays the simulation findings. The findings show that the actual LULC in the Nile Nyabarongo River in 2020 and the simulated LULC were very similar. To validate the model's effectiveness in predicting LULCCs, we first simulated the LULC for the year 2020 by applying transition probabilities and area transition values from the years 1990–2000. We related the simulated results with the classified LULC data for 2020, using kappa variations as a measure of accuracy.

The analysis of the simulation yielded kappa variation estimates of 0.95 for $K_{no}$, 0.93 for $K_{location}$, and 0.62 for $K_{standard}$, indicating a strong level of agreement. This demonstrates the reliability and effectiveness of the model in predicting future LULCCs of the basin. Furthermore, upon visual examination, we observed a relatively close correspondence between the LULC categories in the simulated data for 2020 and the classified data for the same year. Results demonstrated that the simulation results were reliable and that the simulated LULC outline was nearly identical to the real pattern.

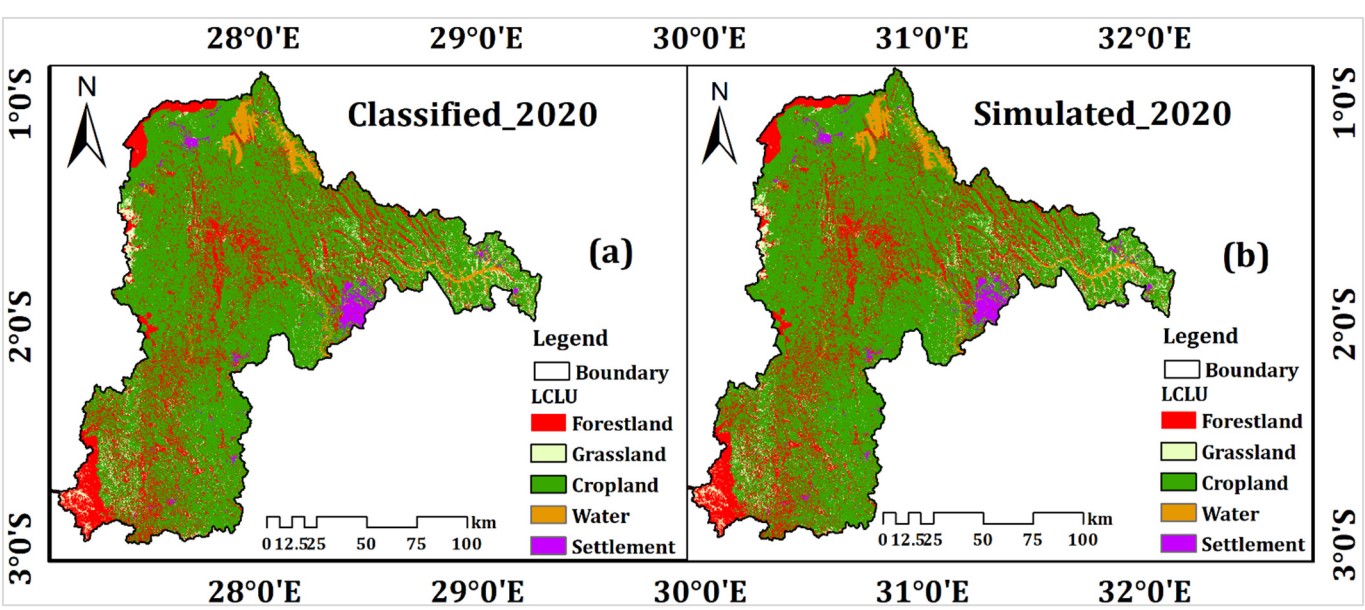

**Figure 4.** Modeled and annotated LULC maps for (**a**) 2020 Classified and (**b**) Simulated 2020.

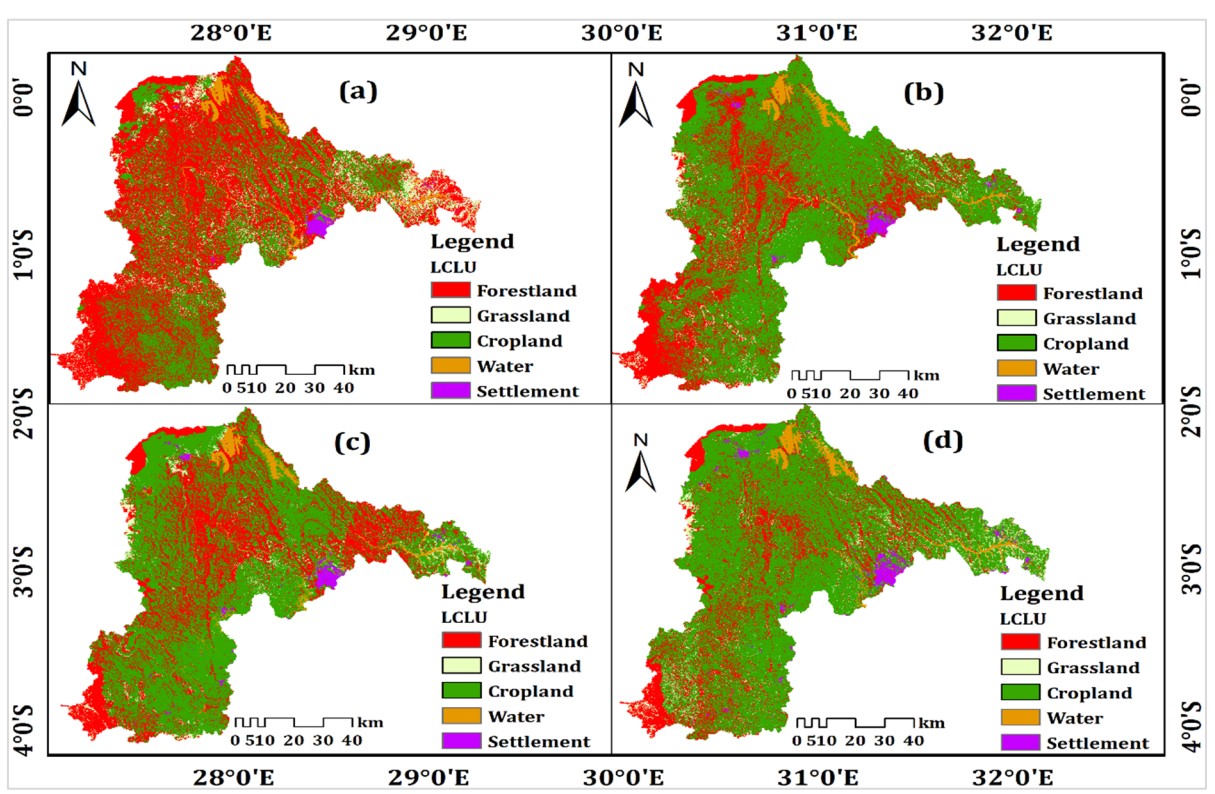

**Figure 5.** Changes of LULC in the basin for (**a**) 1990, (**b**) 2000, (**c**) 2010, and (**d**) 2020.

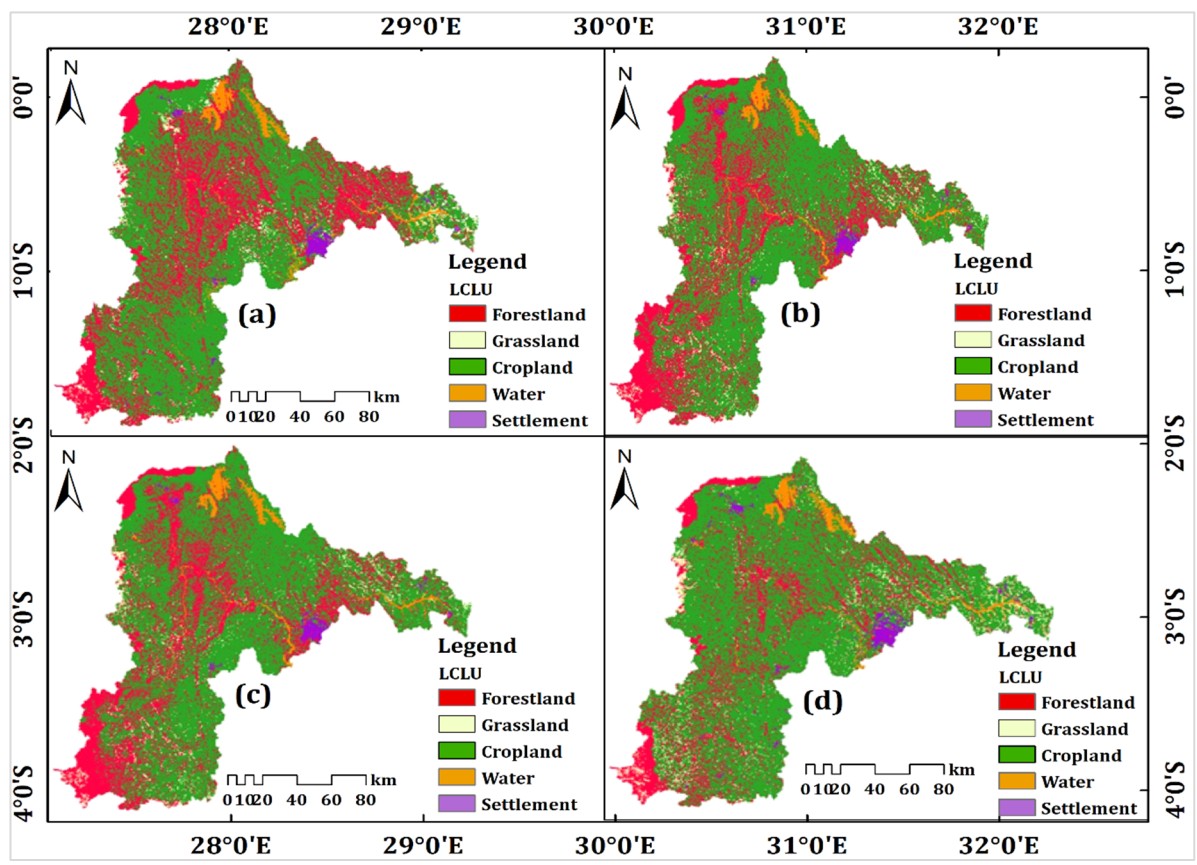

**Figure 6.** Projected LULC for (**a**) 2030, (**b**) 2040, (**c**) 2050, and (**d**) 2060.

## 3. Results and Discussion

### 3.1. Changes of LULC Types in the Basin

The study area indicated five classes of LULC during the time period. The historical changes and the patterns of change in each LULC class were revealed by the evaluation of LULC variation. The major LULC types in the research area include cropland, grassland, and forests, and the overall amount virtually changed between 1990 and 2020. Nevertheless, an examination of the transitions between these land-cover types demonstrated a significant conversion among them, as illustrated in Table 6. The categorized maps' accuracy was evaluated using the overall accuracies and kappa coefficients, which were found to be above 95% and 0.93, respectively (Table 3). This indicates that the classification of images was reliable and accurate for analyzing LULCC. Rwanda has recently encountered various challenges, including urbanization, loss of agricultural land, and a scarcity of water resources [73,74].

**Table 6.** LULC distribution in 1990, 2000, 2010, and 2020 in the catchment.

| No | LCLU Category | 1990 | | 2000 | | 2010 | | 2020 | |
|----|---------------|------|------|------|------|------|------|------|------|
| | | Area/km² | Area/% | Area/km² | Area/% | Area/km | Area/% | Area/km² | Area/% |
| 1 | Forestland | 4694.6 | 58.2 | 2608.1 | 31.7 | 1679.1 | 32.8 | 1072.4 | 17 |
| 2 | Grassland | 496.7 | 7.4 | 242.2 | 3 | 339.1 | 16.2 | 662.1 | 7.7 |
| 3 | Cropland | 2898.9 | 32.2 | 5264.1 | 63 | 5989.6 | 45.8 | 6223.2 | 71.4 |
| 4 | Settlement | 69.5 | 0.7 | 78.5 | 1.1 | 195.1 | 1.2 | 255.3 | 2.6 |
| 5 | Water | 121.4 | 1.5 | 88.2 | 1.2 | 78.2 | 4 | 68.1 | 1.3 |
| | Total | 8281.1 | 100 | 8281.1 | 100 | 8281.1 | 100 | 8281.1 | 100 |

The result shows significant changes in other LULC classes between 1990 and 2020. It indicates a decrease in forestland area from 4694.6 km$^2$ in 1990 to 1072.4 km$^2$ in 2000. This is a decrease from 58.2% to 17% of the total area (Table 6). The conflicts and upheavals in Rwanda during the early 1990s resulted in significant population loss through death and displacement. Subsequently, as the conflicts subsided, the displaced population gradually returned, leading to the recovery of the overall population. Nevertheless, due to the economic repercussions of the war, the successful survival and well-being of the returning population remained uncertain [36]. Numerous acres of forest were thus cleared for firewood and charcoal in order to address the energy shortage brought on by the inadequate production capacity [75]. Grassland decreased from 496.7 km$^2$ in 1990 to 242.2 km in 2000 and increased drastically to 662.1 km$^2$ in 2020. The result indicates an increase from 3% of the total area in 2000 to 7.7% in 2020. The cropland increases from 2898.9 km$^2$ in 1990 to 6223.2 km$^2$ in 2020. This indicates an increase from 32.2% of the total area in 1990 to 71.4% in 2020 [76]. This implies that the crops grown on the restricted cropland must not only fulfill the essential requirements of the extensive population but also contribute to economic growth, aligning with the goals of rapid development. The settlement increased from 69.5 km$^2$ in 1990 to 255.3 km$^2$ in 2020. The result indicates an increase from 0.7% of the total area in 1990 to 2.7% in 2020. The water in the Nile Nyabarongo River basin changes during the entire study period; it decreased from 121.4 km$^2$ (1.5% of the total area) in 1990 to 68.1 km$^2$ (1.3% of the total area) in 2020 (Table 6).

The transition matrix (Table 7) was created and utilized to forecast LULCCs in the years 2030, 2040, 2050, and 2060.

**Table 7.** Transition Probability Matrix calculated using land use in 1990 and 2020.

| Class | Forestland | Grassland | Cropland | Settlement | Water | Total |
|---|---|---|---|---|---|---|
| Forestland | 0.8 | 0.0 | 0.0 | 0.1 | 0.0009 | 0.9 |
| Grassland | 0.0 | 0.7 | 0.0 | 0.0 | 0.007 | 0.7 |
| Cropland | 0.1 | 0.3 | 0.4 | 0.0 | 0.0166 | 0.9 |
| Settlement | 0.0 | 0.0 | 0.0 | 1.0 | 0 | 1.0 |
| water | 0.0 | 0.1 | 0.0 | 0.0 | 1.0 | 1.1 |
| Total | 1.0 | 1.4 | 0.6 | 1.1 | 0.9 | 5.0 |

A study of the Nile Nyabarongo River's LULC pattern from 1990 to 2020 revealed a striking transformation (Table 6). The analysis indicated both minor and major changes across different LULC classes over the past three decades. The four maps illustrate the distinct stages each LULC class has gone through, each exhibiting its own rate and extent of change. Table 6 presents quantitative data, showcasing the statistics of the LULC categories and how they changed over the given periods.

### 3.2. Future LULCC Prediction Results

In the Nile Nyabarongo River basin, changes in LULC over the next 30 years will primarily involve decreased forest area and increased grassland and cropland area, accompanied by the expansion of settlements. This is directly influenced by the population growth in the area. To ensure food security, reclaiming forestland is the most effective approach, considering a particular unit of cropland output [77]. Furthermore, the agricultural system relies heavily on rain-fed farming; the two rainy periods were when most food crops are germinated and processed. However, these seasons experience potent rains and several floods, necessitating the implementation of agricultural drainage systems in the area [78,79]. Nevertheless, the agricultural drainage system in the Nile Nyabarongo River basin is underdeveloped. As a result, the region makes use of the natural advantages offered by the topography to address this issue.

The analysis of the transition probability maps indicated significant shifts, specifically the conversion of forestland and water into cropland, which implies forest disturbance. These changes were considered when predicting future transformations in 2030, 2040, 2050,

and 2060 (Figure 6). In accordance with the ongoing changes, we designed the changes in LULC and got a map of changes after 30 years. From the figures (Figure 6), it can be observed how changes will mainly happen in cropland and settlement, and how their territory is expanding. In addition, the expansion extends to grassland, but currently the area of forests is significantly reduced. The outcome of this study confirmed ample proofs of transition in LULC. As we have mentioned, the main change at LULC is superior to cropland and settlement areas and to grassland. Here (Figure 6), you can see that all the changes relate to human factors; an increase in the population indicates that the demand for resources is increasing.

According to the simulation results of the CA–Markov model (Table 8), it is predicted that the settlement area will experience significant growth over time. Specifically, the settlement area is expected to increase from 310.5 km$^2$ in the year 2030 to 453.2 km$^2$ in the year 2060. In terms of cropland cover, there will be a gradual increase from 6292.1 km$^2$ in 2030; the cropland cover is projected to reach about 6420.6 km$^2$ by 2060. On the other hand, the area of forestland is anticipated to decrease during the same period. Starting from 920.1 km$^2$ in 2030, the forestland area is expected to decline to approximately 508.2 km$^2$ in 2060. As for grassland, there will be a slight increase in its area. Specifically, the grassland area will grow from 699.1 km$^2$ in 2030 to 859.1 km$^2$ in 2060. Lastly, water bodies are expected to undergo a slight reduction in size. The water area is predicted to decrease from approximately 59.2 km$^2$ in 2030 to around 40 km$^2$ in 2060.

**Table 8.** Statistical distribution of the modeled LULC in 2020, 2030, 2034, 2050, and 2060.

| No | LCLU Category | 2020 | | 2030 | | 2040 | | 2050 | | 2060 | |
|---|---|---|---|---|---|---|---|---|---|---|---|
| | | Area/km$^2$ | Area/% | Area/km$^2$ | Area/% | Area/km$^2$ | Area/% | Area/km$^2$ | Area/% | Area/km$^2$ | Area/% |
| 1 | Forestland | 1072.4 | 58.2 | 920.1 | 31.7 | 801.7 | 30.1 | 771.6 | 16.9 | 508.2 | 31.3 |
| 2 | Grassland | 662.1 | 7.4 | 699.2 | 3 | 761.1 | 17.2 | 802.4 | 7.8 | 859.1 | 20.3 |
| 3 | Cropland | 6223.2 | 32.2 | 6292.1 | 63 | 6307.2 | 47.5 | 6249.2 | 71.4 | 6420.6 | 45.9 |
| 4 | Settlement | 255.3 | 0.7 | 310.5 | 1.1 | 355.2 | 1.2 | 407.4 | 2.6 | 453.2 | 1.4 |
| 5 | Water | 68.1 | 1.5 | 59.2 | 1.2 | 55.9 | 4 | 50.5 | 1.3 | 40 | 1.1 |
| | Total | 8281.1 | 100 | 8281.1 | 100 | 8281.1 | 100 | 8281.1 | 100 | 8281.1 | 100 |

These findings highlight the potential changes in LULC in the study area, as simulated by the CA–Markov model. These findings imply that the growth of settlements will play a crucial role in decreasing non-cropland areas, exposed soils, landfills, and excavation sites. The presence of infrastructure and educational and medical facilities in the region will also lead to a significant rise in the urban population and the transformation of land utilization, impacting the natural environment. Research conducted in China has highlighted that the expansion of grassland, cropland, and settlements and, especially, cropland expansion and rapid urbanization are among the key drivers behind extensive construction activities [80–82]. Without controlling urban development, this trend will diminish the country's agricultural land and hinder sustainable urban expansion goals. Therefore, it is necessary to revise certain policies in the Nile Nyabarongo River basin that promote rapid urban growth without considering the harmonious coexistence of human activities and the ecological environment. By doing so, local governments and city planners are capable of controlling land use and understanding the complicated processes of development while preserving the natural environment.

The CA–Markov model showed great efficiency in modeling the geospatial techniques in LULCC in the area, implying that the area is prone to change and requires extensive land-use planning and water resource management. Past LULC changes, used together with LULC drivers/factors, have shown their applicability in simulating future LULCC [83–85], as observed, and should be documented for future reference. However, since this study was limited in the use of driving factors, and because future LULC patterns are uncertain, factors such as population growth, climate variability, natural hazards, and socio-economic

developments should also be considered in order to enhance our understanding of LULC change processes.

*3.3. Drivers of LULCCs*

Generally, changes in LULC occur because of a multitude of complex and diverse factors [65,86]. Previous research indicates that human activities are the primary driver of LULCCs on a global scale. However, the specific factors driving these changes may vary depending on the characteristics of the area [86]. In our study, we conducted an analysis to identify the various driving factors contributing to LULCCs in our specific study area. Results indicate that the forest, grassland, and cropland areas have undergone changes over time, and the factors responsible for these changes have also gradually evolved. In the Nile Nyabarongo River basin, multiple factors influenced the changes in forest, grassland, and cropland areas, while changes in water resources and settlements were predominantly driven by a single factor compared to the previous three. In the East Africa region, population growth emerged as the primary driver of land-use and land-cover (LULC) changes, but this was not the case for forest change [87]. This is due to the ongoing transformation of low-altitude forests into agriculture land, while the newly added forest areas were mostly located in high-altitude regions unsuitable for human settlement or in protected areas like the Nyungwe Forest National Park. In these sparsely populated areas, population pressure on land use could not have a substantial impact.

Moreover, the Nile Nyabarongo River basin experiences a moderate tropical plateau environment that receives lots of rain and encourages rapid plant development. As a result, precipitation emerged as the primary catalyst for change. Notably, the population played a crucial role in driving alterations in grassland, while topography acted as a constraint on its expansion. The transformation of grasslands was predominantly witnessed in locations with plenty of rainfall and a substantial population; precipitation and population are the main driving reasons behind these alterations. Consequently, the driving factor for the alteration in water areas shifted from topography–population to location–topography. The natural rivers and lakes that make up the majority of the water bodies in the Nile Nyabarongo River basin are essential water sources for a variety of human activities. Therefore, the population has an impact on how the water area changes [88]. The cropland in the river valleys plays a significant role in the overall cropland in the area. The Nile Nyabarongo River basin experiences plenty of rain, particularly during the rainy season, leading to higher water levels in its rivers. However, due to population growth and increased urbanization, there has been an upsurge in the population's need for water. As a result, most of the cropland in the river valleys has been converted to water storage or river channels. The main driving force behind the settlement has shifted from population to the quality of the soil. While population was previously the primary factor driving urbanization, soil has become a secondary factor as new cities are now planned more scientifically, taking into consideration factors beyond just population. The properties of the soil are crucial in the development of cropland.

## 4. Conclusions

The goals of this research were to evaluate and predict changes in LULC in the Nile Nyabarongo River basin. To achieve this, a unified hybrid CA–Markov model, cellular automata, and Markov chains were utilized. GIS methods and multitemporal satellite data were used to observe current LULCCs from 1990 to 2020. Through a review of LULC transfer in the Nile Nyabarongo River during the span of the historical period (1990–2020), this study discovered that the development of agriculture and the decline of forested land accounted for the majority of the LULC in the basin. Moreover, water varied as grassland and settlement increased annually. Although changes in water and settlement were mostly impacted by one factor, frequent factors contributed to changes in the forest, grassland, and agricultural sectors.

The LULC maps were divided into five categories: forestland, grassland, cropland, settlement, and water. This thorough comprehension of the basin's LULCCs allowed for future scenario predictions. Using kappa empirical indicators, the reliability of the simulated LULC map for 2020 was assessed, which showed a significant correlation with the map obtained from satellite data, indicating the reliability of the simulation model. The variables influencing various LULC types also evolved throughout time. The LULC pattern in the Nile Nyabarongo River for the following 30 years was simulated using the CA–Markov model. Simulation findings demonstrated that the CA–Markov model provides an accurate LULC pattern and has an excellent simulation accuracy. In the Nile Nyabarongo River, the LULC will continue to indicate an increase toward less forest and more grassland and crops. However, it is important to note that while the study successfully predicted the expected LULCCs for the future, it omitted the most recent census statistics on local population growth. To fully understand the processes of long-term LULCCs and how they affect the future of sustainability, additional study should investigate the relationship between population increase and land-cover alterations.

**Author Contributions:** Methodology, A.G.; Software, A.G.; Validation, A.G.; Formal analysis, B.R.H. and U.D.E.; Data curation, U.D.E.; Visualization, A.G.; Funding acquisition, C.X. All authors have read and agreed to the published version of the manuscript.

**Funding:** This study was funded and supported by the Pan-Third Pole Environment Study for a Green Silk Road Strategic Priority Research Program of the Chinese Academy of Sciences (Grant No. XDA20060303), the Key Program of the National Natural Science Foundation of China (42230708), the K. C. Wong Education Foundation (GJTD-2020-14), the Chinese Academy of Sciences President's International Fellowship Initiative (PIFI, Grant No. 2017VCA0002) and the National Natural Science Foundation of China (32071655).

**Data Availability Statement:** The source of all the data used in this study is provided in the manuscript.

**Acknowledgments:** The authors would like to express their gratitude to the Chinese Academy of Science (UCAS) for their support in conducting this research. Additionally, the authors greatly acknowledge the support received from the Xinjiang Institute of Ecology and Geography, Chinese Academy of Sciences (CAS). Various funding sources have contributed to this study, including the Pan-Third Pole Environment Study for a Green Silk Road Strategic Priority Research Program of the Chinese Academy of Sciences (Grant No. XDA20060303) and the Key Program of the National Natural Science Foundation of China (42230708). Furthermore, the K. C. Wong Education Foundation (GJTD-2020-14), the Chinese Academy of Sciences President's International Fellowship Initiative (PIFI, Grant No. 2017VCA0002), and the National Natural Science Foundation of China (32071655) have all collaborated to fund this research.

**Conflicts of Interest:** The researchers have fully complied with ethical considerations regarding conflicts of interest and have stated that there are no conflict of interest to disclose.

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
