# Peer review of "A CA–Markov-Based Simulation and Prediction of LULC Changes over the Nyabarongo River Basin, Rwanda"

_land, doi:10.3390/land12091788_

Round 1

Author Response

The authors have diligently worked on the revision as suggested by the first reviewer and have made substantial improvements to the manuscript. It is our sincere hope that the revisions we have made will address any concerns or issues raised during the initial review. we are eager to receive feedback on the revised version, as it would guide us further in enhancing the quality and impact of my work.

Reviewer 2 Report

Review on land-2586046

General comment

The authors aimed to evaluate the changes in LULC from 1990 to 2020 16 and predict future fluctuations until 2060 using one of the well-known spatiotemporal prediction models of LULC (i.e., CA-Markov). The model was applied, the scientific contribution of the manuscript is minimal, however, for the study area it would be of great value for local management and development.

There are huge mismatching between the classified and the simulated LULC maps for the year 2020 especially in the Forest class (Figure 4). This could rise a major concern in the input maps to the model.

Specific Comments

Table 1. RS… à Landsat….

·        In table 1: The resolution of Landsat 5 is 30m not 60

·        The images were selected from different months (i.e., July and September) this could make problems in the analysis.  

2.2.2. Evaluation of classification accuracy:

·        Please specify the number samples and the type of sampling strategy (e.g.,  stratified random sampling…… etc.)

Lines 313-314 dose not match with information in Table 4!. The authors stated that “cropland, grass-313 land, and forests, and the overall amount remained virtually unchanged between 1990 314 and 2020.” While there are huge change in the LULC types in the table.

Line 334-353 are quit confusing and don’t match the results in table 1. Please clarify.

Figure 5: if (d) is the 2020 then it does not match (a) in Figure 4.

In figure 6: the predicted map of 2030 is bizarre! It seems to be wrong comparing to the input maps to the model.

Figure 6 is not clear in general, and it should have the same color scheme as in figure 5.

Line 406: reforestation!! The results showed decreasing in the Forst land even in the predicted maps.

The authors are recommended to revise their discussion of the results to match with the numbers in the tables. They should make sure about the accuracy of the input LULC maps.

Author Response

The authors have diligently worked on the revision as suggested by the second reviewer and have made substantial improvements to the manuscript. It is our sincere hope that the revisions we have made will address any concerns or issues raised during the initial review. we are eager to receive feedback on the revised version, as it would guide us further in enhancing the quality and impact of my work.

Round 2

Reviewer 1 Report

The authors have addressed my concerns nicely.

Best wishes,

Author Response

The authors thank you very much.

Reviewer 2 Report

Please make Figure 6 to have the same colors of the classes in figure 5.

In the caption of the figure add (a), (b)....... 

Author Response

(The authors gave the same response as above.)
